# Increasing Mass Timber Consumption in the U.S. and Sustainable Timber Supply

**Jeff Comnick [1], Luke Rogers [1] and Kent Wheiler [2,*]**

1   Natural Resource Spatial Informatics Group, University of Washington, Seattle, WA 98195, USA; jcomnick@uw.edu (J.C.); lwrogers@uw.edu (L.R.)
2   Center for International Trade in Forest Products, University of Washington, Seattle, WA 98195, USA
*   Correspondence: kwheiler@uw.edu; Tel.: +1-253-218-8872

**Abstract:** Mass timber products are growing in popularity as a substitute for steel and concrete, reducing embodied carbon in the built environment. This trend has raised questions about the sustainability of the U.S. timber supply. Our research addresses concerns that rising demand for mass timber products may result in unsustainable levels of harvesting in coniferous forests in the United States. Using U.S. Department of Agriculture U.S. Forest Service Forest Inventory and Analysis (FIA) data, incremental U.S. softwood (coniferous) timber harvests were projected to supply a high-volume estimate of mass timber and dimensional lumber consumption in 2035. Growth in reserve forests and riparian zones was excluded, and low confidence intervals were used for timber growth estimates, compared with high confidence intervals for harvest and consumption estimates. Results were considered for the U.S. in total and by three geographic regions (North, South, and West). In total, forest inventory growth in America exceeds timber harvests including incremental mass timber volumes. Even the most optimistic projections of mass timber growth will not exceed the lowest expected annual increases in the nation's harvestable coniferous timber inventory.

**Keywords:** mass timber; cross-laminated timber (CLT); embodied carbon; sustainable timber supply; forest inventory; reforestation; seedlings; replanting

## 1. Introduction

Mass timber, short for massive timber, is a category of wood products that are engineered for use in large structural applications. Mass timber is made using solid sawn lumber or veneer to create large wood panels, columns, or beams for load-bearing walls, floor, and roof construction. The most common mass timber product is cross-laminated timber (CLT), but there are several variations such as dowel-laminated timber (DLT), nail-laminated timber (NLT), mass plywood panels (MPP), etc. Mass timber products can be made from both softwoods and hardwoods of many different species if engineered properly, but construction-grade softwoods such as spruce, pine, and fir are commonly employed. Mass timber is engineered for high strength ratings like concrete and steel but is usually lighter in weight and can be prefabricated and delivered to the job site, facilitating faster construction with less labor, material waste, and noise pollution on site. Substituting mass timber for concrete and steel avoids the carbon emissions related to the energy-intensive manufacturing of those materials and helps reduce embodied carbon in the built environment. Mass timber products were first developed in Austria in the 1990s, and are widely used throughout Europe, but are now growing in popularity across the United States. As of September 2021, 1241 mass timber projects had been constructed or were in design in the multi-family, commercial, or institutional categories. This total includes modern mass timber and post-and-beam structures built since 2013 [1]. News stories extolling the benefits of mass timber are now common, recently including CNN [2], Popular Mechanics [3], Vox [4], and the New York Times [5]. The option to build high-rise

structures with wood has been embraced by the architecture community as they pursue their commitment to design carbon-neutral buildings [6].

As is well known, trees absorb carbon dioxide and release oxygen, effectively helping to mitigate climate change. The forest sector can help reduce atmospheric carbon dioxide ($CO_2$) by sequestering and storing carbon in trees and wood products and producing energy and materials that use relatively less fossil fuel compared to functional alternatives [7]. Research has highlighted that after trees have been harvested and turned into lumber, homes, furniture, etc., those wood products and structures continue to keep a large proportion of the tree carbon in a sequestered form [8], while replanted trees absorb more carbon as they grow [9]. This benefit of using wood for long-term structural applications is compounded when substitution for energy-intensive concrete and steel is considered. As a result, wood products, and particularly structural wood products such as mass timber, play an important role in combating climate change—one that has often been ignored in climate mitigation assessments.

The growing popularity of mass timber has raised questions among architects and designers regarding the sustainability of the U.S. timber supply. They are interested in the carbon benefits of substituting wood for concrete and steel in structural applications but do not want their material choice to lead to deforestation. Accordingly, this research addresses the question: will rising demand for mass timber products result in unsustainable levels of harvesting in coniferous forests in the United States?

In 2000, the North American Forest Commission of the Food and Agriculture Organization (FAO) reported that in the United States "forest growth nationally has exceeded harvest since the 1940s. By 1997 forest growth exceeded harvest by 42% and the volume of forest growth was 380% greater than it had been in 1920" [10].

In 2016 the National Alliance of Forest Owners (NAFO) commissioned Forest2Market (https://www.forest2market.com) (accessed on 1 November 2021) to determine, using Forest Inventory and Analysis (FIA) data (https://www.fia.fs.fed.us) (accessed on 1 November 2021) from the U.S. Forest Service (USFS) if privately owned timberlands are growing more wood than is being removed. It is a long- and well-established fact that private forests supply the vast majority of wood harvested in the U.S. [10]. FIA data has been collected since 1930 and is the most comprehensive inventory and analysis of the present and prospective conditions of U.S. forests [11]. Using this data, Forest2Markets found that between 2008 and 2015, 49 percent more conifers were grown than were harvested [12].

Forest inventory, particularly on private land, depends on replanting after harvest. While inventory is an important static measure, replanting tells a dynamic message about the future of our forests. Since 2012 the USFS has released an annual report, Forest Nursery Seedling Production in the United States. Data are collected for all fifty states and organized by nine regions. It includes conifer seedlings produced and conifer seedlings imported from Canada. It is the most complete compilation of such data in the country. For 2020 they report that forest nursery production was more than 1.25 billion tree seedlings, including about 18.5 million container seedlings imported from Canada, and 97% of seedlings produced were softwood species [13].

While forest inventories have been increasing and a high level of replanting occurs, mass timber construction is a recent development creating new incremental demand on the nation's wood supply. In 2020 the Softwood Lumber Board (SLB), a USDA checkoff program, working with FPInnovations and Romanchych Consulting projected incremental softwood lumber consumption reflecting the growing use of mass timber [14]. Lumber consumption in 2035 was estimated under four scenarios (Base Case, Low-, Medium-, and High-Volume Cases). Incremental consumption estimates are available regionally (North, South, and West). Their projections, in million board feet, are shown in Table 1.

**Table 1.** Projected incremental softwood lumber consumption in 2035 (million board feet, MMBF).

| Region | Base Case | Low Volume | Medium Volume | High Volume |
|---|---|---|---|---|
| North | 2290 | 1761 | 2484 | 3155 |
| South | 1784 | 1372 | 1930 | 2445 |
| West | 838 | 645 | 906 | 1142 |
| Total | 4912 | 3778 | 5320 | 6742 |

Using the SLB projections, our research examines long-term softwood timber supply in the United States considering the growing use of wood products and in particular mass timber products. To be conservative, we used the High-Volume estimates for our analysis.

## 2. Materials and Methods

Forest growth and removals (harvests) for the United States were summarized using the rFIA package [15] for the R programming language [16]. This package facilitates downloading and summarizing FIA datasets [17]. FIA reports merchantable board foot volume using the International 1/4-inch Log Rule for all softwood trees greater than 9 inches in diameter and all hardwood trees greater than 11 inches in diameter at breast height. Population estimates and 95% confidence intervals were produced at the regional (North, South, and West), state, and county levels (see Table 2 for a list of states by region); and by owner class, public reserve status, and proximity to water. We used the proximity to water attribute in FIA to partition growth by riparian (near water) and upland (not near water) management zones.

**Table 2.** List of states by region.

| Region | States |
|---|---|
| North | CT, DE, IA, IL, IN, MA, ME, MI, MN, MO, NY, NH, NJ, OH, PA, RI, VT, WI, WV |
| South | AL, AR, FL, GA, KT, LA, MD, MS, NC, OK, SC, TN, TX, VA |
| West | AZ, CA, CO, ID, KS, MT, ND, NE, NM, NV, OR, SD, UT, WA, WY |

Regional lumber consumption was apportioned to production using current FIA harvest estimates (Table 3). For example, the North region is estimated to account for 47% of projected mass timber demand but produces only 6% of current harvest volume; after apportioning, harvest volumes in the North region increase by amounts corresponding to 6% of incremental demand.

**Table 3.** Percentage consumption vs. production by region (MMBF).

| Region | Incremental Consumption | Percent | FIA Harvest Volume | Percent | Adjusted Incremental Consumption |
|---|---|---|---|---|---|
| North | 2290 | 47% | 2007 | 6% | 278 |
| South | 1784 | 36% | 20,391 | 57% | 2822 |
| West | 838 | 17% | 13,098 | 37% | 1813 |
| Total | 4912 | | 35,496 | | 4912 |

To assess mass timber demand in the context of current growth and harvest rates, we converted projected consumption values in lumber board feet into log board feet (International 1/4-inch Log Rule used by FIA). Conversion factors are listed in Table 4.

**Table 4.** Factors to convert from lumber board foot volume to International 1/4 board foot volume.

| Region | Recovery Factor | Prop of Sawlog Used for Lumber | Breakage and Defect | Scribner to International $\frac{1}{4}$ BF |
|---|---|---|---|---|
| North | * | 0.28 | 0.9 | 1.4 |
| South | * | 0.48 | 0.9 | 1.13 |
| West | * | 0.36 | 0.9 | 1.1 |

Note: * Proprietary recovery factors provided by an industry consulting organization which they derive based on their extensive analysis of sawmill production data. Their numbers are confidential but not dissimilar to recovery factors common in the industry ranging between 1.7 and 2.5.

Lumber board foot volume was converted into Scribner board foot log volume using proprietary recovery factors that are not dissimilar to other recovery factors common in the industry ranging between 1.7 and 2.5. The Scribner board foot log volume was adjusted by the proportion of sawlog volume used for lumber. We derived this factor for each region by comparing current FIA harvest volume and lumber production estimates provided in Howard and Liang [18]. We then adjusted log volume for breakage and defect (assuming a 10% breakage and defect value) and converted from Scribner to International 1/4 volume. We then calculated the average value for the Scribner/International 1/4 ratio for each region. Summarized results regionally and for the entire United States are shown in Table 5 (Base Case) and Table 6 (High-Volume Case).

**Table 5.** Conversion of lumber board foot volume to International 1/4 board foot volume—Base Case.

| Conversion Steps | North | South | West | Total |
|---|---|---|---|---|
| Lumber BF | 278 | 2822 | 1813 | 4912 |
| Lumber BF to Scribner BF | 155 | 1576 | 1013 | 2744 |
| Adjust by % of Sawlog Used for Lumber | 323 | 3284 | 2110 | 5717 |
| Adjust for Breakage and Defect | 359 | 3649 | 2344 | 6352 |
| Scribner BF to International 1/4 BF | 318 | 3229 | 2074 | 5621 |

**Table 6.** Conversion of lumber board foot volume to International 1/4 board foot volume—High-Volume Case.

| Conversion Steps | North | South | West | Total |
|---|---|---|---|---|
| Lumber BF | 381 | 3873 | 2488 | 6742 |
| Lumber BF to Scribner BF | 213 | 2164 | 1390 | 3767 |
| Adjust by % of Sawlog Used for Lumber | 444 | 4508 | 2895 | 7847 |
| Adjust for Breakage and Defect | 493 | 5009 | 3217 | 8719 |
| Scribner BF to International 1/4 BF | 436 | 4432 | 2847 | 7716 |

To be conservative, this analysis compared the lower end of the confidence interval for net growth with the upper end of the confidence interval for harvesting and the Volume High mass timber scenario. We first compared projected mass timber demand to current lumber consumption and harvest volume rates. Current harvest with and without mass timber demand is then compared to current net growth. We also break out net growth by upland (not near water) and riparian (near water) zone and compare the upland growth only with harvest volumes with and without mass timber demand. This is important particularly in the West region where Forest Practice Acts in Washington, Oregon, and California limit riparian harvesting.

## 3. Results

Incremental conifer lumber consumed if the mass timber forecast volumes are achieved relative to total current annual lumber consumption is shown in Figure 1 and Table 7.

Note that lumber volumes are in terms of billion board feet (BBF). Average consumption between 2010 and 2017 was calculated from Table 28 in Howard and Liang [17]. Regional consumption was estimated by multiplying average consumption for the United States by the regional proportion of projected mass timber demand for the High-Volume scenario in 2035. Average consumption is 39.5 BBF with incremental mass timber consumption adding a 17% increase in consumption.

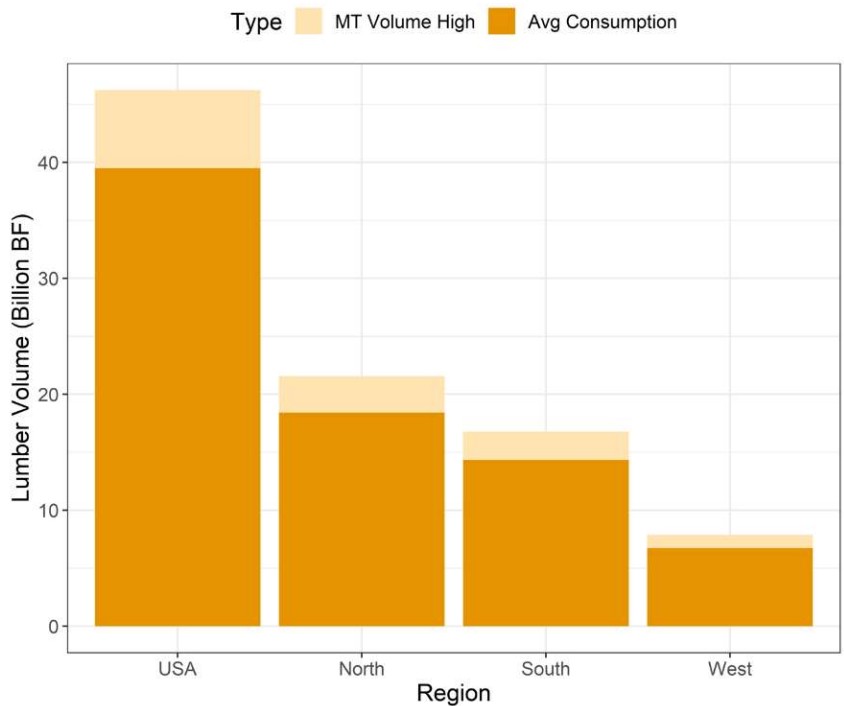

**Figure 1.** Average lumber consumption (2010–2017) and projected mass timber demand—High-Volume 2035 Case.

**Table 7.** Average lumber consumption (2010–2017) and projected mass timber demand—High-Volume 2035 Case.

| | Billion Board Feet (BBF) | | | | |
|---|---|---|---|---|---|
| | USA | | North | South | West |
| | BBF | Percent Growth | BBF | BBF | BBF |
| Average Lumber Consumption | 39.5 | | 18.4 | 14.3 | 6.7 |
| Projected Mass Timber Demand (High Volume 2035) | 6.7 | 17% | 3.2 | 2.4 | 1.1 |
| | 46.2 | | 21.6 | 16.8 | 7.9 |

Having estimated consumption, we then look at harvests necessary to support that consumption. Incremental harvest volumes if the mass timber forecast volumes are achieved relative to current harvest volumes are shown in Figure 2 and Table 8. Mass timber demand in terms of lumber board feet was converted to harvest volume board feet using the International 1/4 log rule and then apportioned to the region of production.

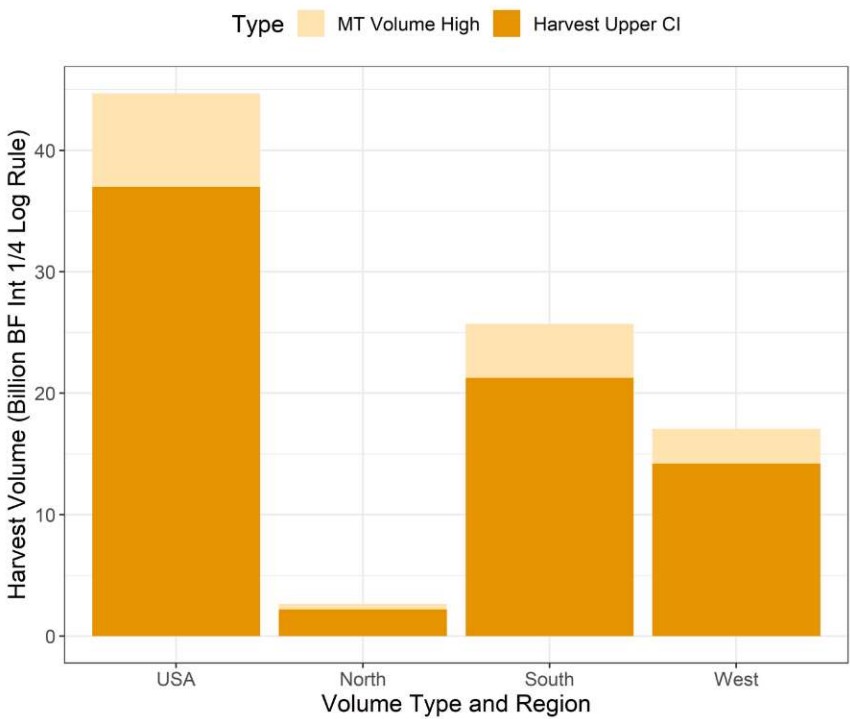

**Figure 2.** FIA estimated harvest volume with projected mass timber demand—High-Volume 2035 Case.

**Table 8.** FIA estimated harvest volume with projected mass timber demand—High-Volume 2035 Case.

| | Billion Board Feet (BBF) | | | | |
|---|---|---|---|---|---|
| | **USA** | | **North** | **South** | **West** |
| | **BBF** | **Percent Growth** | **BBF** | **BBF** | **BBF** |
| Average Lumber Consumption Projected Mass Timber Demand (High Volume 2035) | 37.0 | | 2.2 | 21.3 | 14.2 |
| | 7.7 | 21% | 0.4 | 4.4 | 2.8 |
| | 44.7 | | 2.6 | 25.7 | 17.1 |

The average FIA timber harvest volume is 37 BBF, and the projected mass timber demand adds another 7.7 BBF to harvest levels, an increase of 20.9% in required harvest volumes. You may notice that consumption shown in Figure 1 is more than the harvest volume shown in Figure 2, but that is simply a factor of the different scaling methods used for logs and lumber. Log volume board feet according to the International $\frac{1}{4}$ log rule is measured from the small end diameter of the log and is usually less than the board feet of lumber that can actually be produced, a phenomenon referred to as "overrun".

With estimated harvest volumes on a comparable scale, we can now look at both the current rate of harvesting and the added mass timber harvesting relative to growth and reforestation. Figure 3 and Table 9 show growth compared with current harvesting—harvesting is 56% of growth—and Figure 4 and Table 10 show the same growth numbers (the lowest estimates of growth) compared with the harvesting required to meet current demand plus the added high-volume estimate of mass timber consumption. In the latter scenario, growth still exceeds harvesting by 32%.

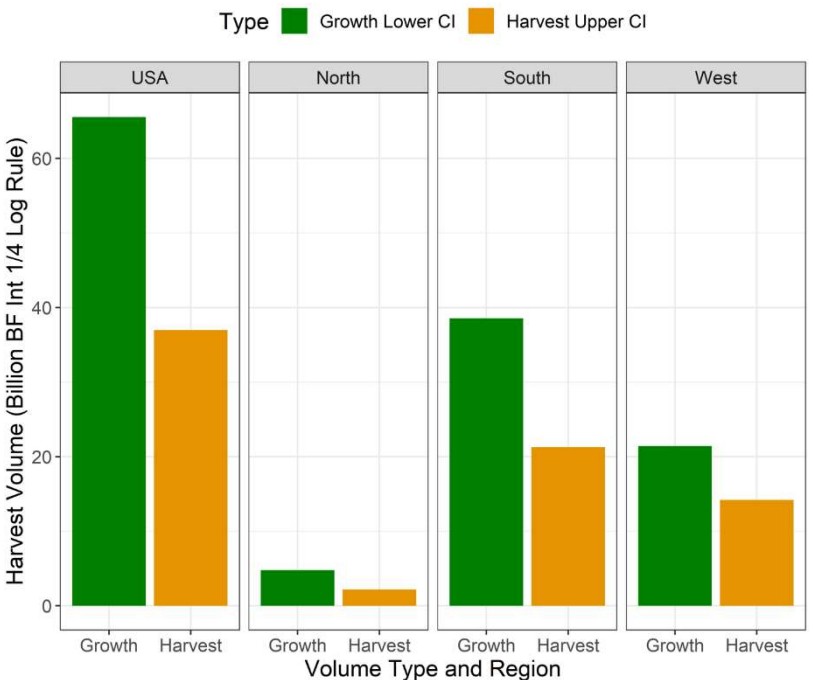

**Figure 3.** FIA estimated growth and harvest volume.

**Table 9.** FIA estimated growth and harvest volume.

| | Billion Board Feet (BBF) | | | | | | | |
| | USA | | North | | South | | West | |
| | BBF | Percent of Growth | BBF | Percent of Growth | BBF | Percent of Growth | BBF | Percent of Growth |
|---|---|---|---|---|---|---|---|---|
| Estimated Growth (Lower Confidence Interval) | 65.5 | | 4.8 | | 38.5 | | 21.4 | |
| Estimated Harvest (Upper Confidence Interval) | 37.0 | 56% | 2.2 | 46% | 21.3 | 55% | 14.2 | 66% |

As noted earlier, we estimate that current harvests consume 56% of net growth. We grow more timber than we consume in total and in all three regions.

Figure 4 and Table 10 show a comparison of the current rate of harvesting plus incremental mass timber consumption relative to growth and reforestation. Still, growth exceeds harvesting in total (by 32%) and in all regions.

Because our focus is assuring sustainable levels of growth to support incremental demand, as noted earlier we are taking a conservative approach by comparing the lower end of the confidence interval for growth with the upper end of the confidence interval for harvesting, and while also using the High-Volume mass timber scenario. In addition, we also discount FIA growth data for growth occurring in reserve areas on public land (protected public forests, preserves, and conservation areas) and for riparian zones on both public and private land that cannot be harvested due to proximity to bodies of water. Although growth may occur, if it cannot be harvested, it cannot be used to meet demand.

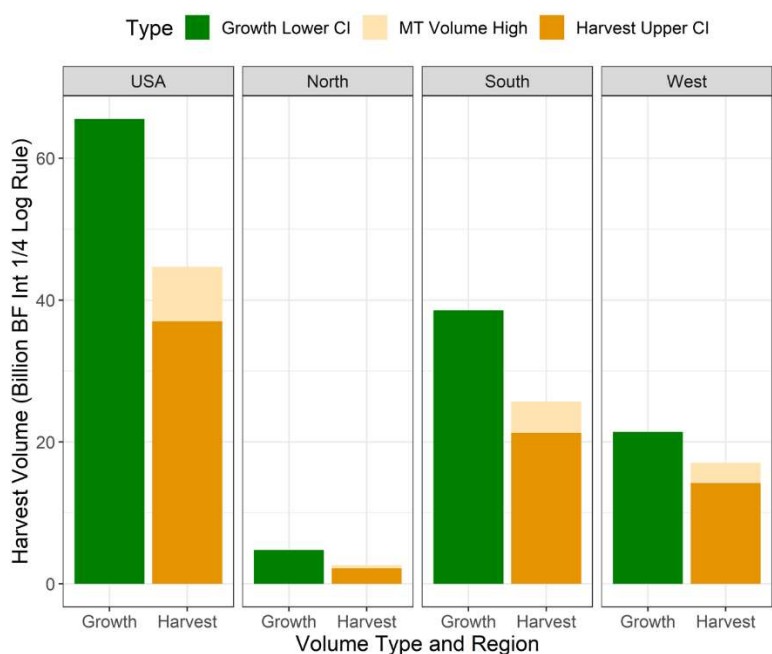

**Figure 4.** FIA estimated growth and harvest volume with projected mass timber demand—High-Volume 2035 Case.

**Table 10.** FIA estimated growth and harvest volume with projected mass timber demand.

| | Billion Board Feet (BBF) | | | | | | | |
|---|---|---|---|---|---|---|---|---|
| | USA | | North | | South | | West | |
| | BBF | Percent of Growth | BBF | Percent of Growth | BBF | Percent of Growth | BBF | Percent of Growth |
| Estimated Growth (Lower Confidence Interval) | 65.5 | | 4.8 | | 38.5 | | 21.4 | |
| Estimated Harvest (Upper Confidence Interval) | 37.0 | 56% | 2.2 | 46% | 21.3 | 55% | 14.2 | 66% |
| Incremental Mass Timber Volume (High Volume) | 7.7 | 12% | 0.4 | 9% | 4.4 | 11% | 2.8 | 13% |
| Excess Growth | 20.8 | 32% | 2.2 | 45% | 12.8 | 33% | 4.4 | 20% |

Reserve status for public land is identified in FIA data. Figure 5 and Table 11 show the proportion of acres by reserve and non-reserve status, and the proportion of growth by the same. On the left side of Figure 5, note that reserves are a relatively small portion of total acres except in the West, where they are 18% of the total area. However, as shown on the right side of the figure, the proportion of growth is tiny in all regions and indiscernible in the West. This may be surprising, but net growth on reserved and protected public land is minimal due to mortality within mature stands and natural disturbances such as wildfire and pine beetle or other pestilences (i.e., growth is offset by natural losses). Without human intervention, a forest does not grow forever. Net growth approaches zero as new growth is offset by natural mortality and losses from natural disturbances.

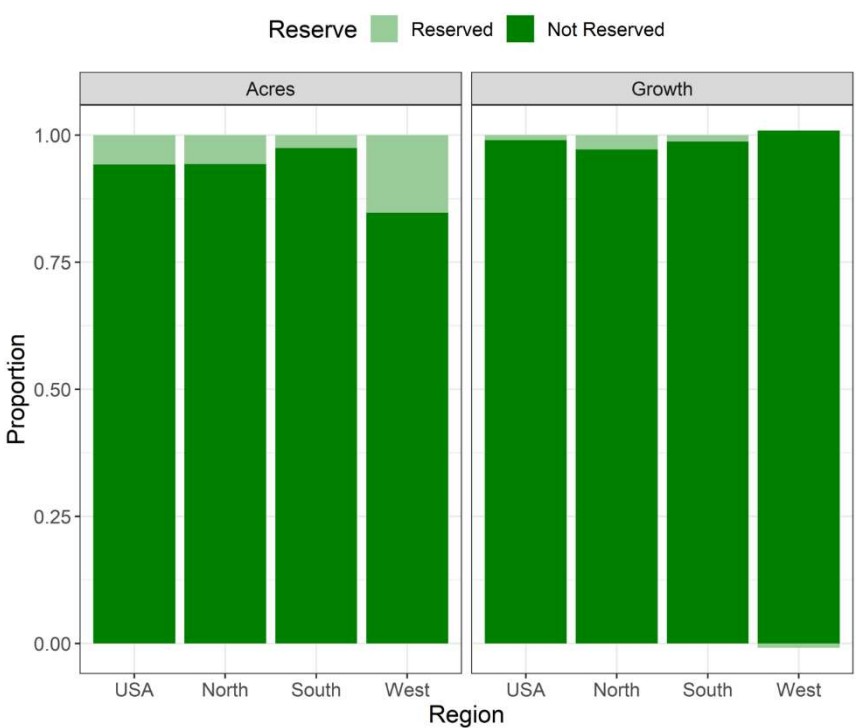

**Figure 5.** Proportion of FIA forest acres and growth volume by reserve status.

**Table 11.** Proportion of FIA forest acres and growth volume by reserve status.

| | Acres | | | | Growth Volume | | | |
|---|---|---|---|---|---|---|---|---|
| | USA | North | South | West | USA | North | South | West |
| Not Reserved | 94.2% | 94.3% | 97.5% | 84.7% | 99.0% | 97.2% | 98.8% | 100.9% |
| Reserved | 5.8% | 5.7% | 2.5% | 15.3% | 1.0% | 2.8% | 1.2% | −0.9% |

Harvesting in riparian zones is allowed in the North and South regions, subject to best management practices, but is restricted by law in Washington, Oregon, and California. FIA does not include a designated category called "Riparian" but does include a measure of "proximity to water" which we used to approximate growth in riparian zones in the West. There are more accurate measures of the riparian area in the three states noted, by county and owner, but without accompanying growth estimates. Were that data to be used, total growth would have to be partitioned proportional to area. We concluded that excluding FIA growth based on proximity to water would result in a more accurate discount. Using this approach, Figure 6 and Table 12 show the proportion of riparian and non-riparian (upland) acres and growth in all regions. Note that growth in riparian zones in the West represents about 20% of total growth and, consequently, potential harvest in the West should be discounted accordingly.

A comparison of the current rate of harvesting relative to growth and reforestation by management zones (riparian vs. upland) is shown in Figure 7 and Table 13. To be clear, Figure 7 is like Figure 3 but with growth split by riparian or upland (non-riparian). Figure 7 begins with the growth and harvest data used in Figure 3 but excludes growth occurring in riparian zones where harvesting is often prohibited. Figure 7 shows that even if we eliminate all growth in riparian zones, we still have enough growth in upland management zones to support current rates of harvesting. (Note that bisecting the data by zone increases the standard error, and because we report the lower confidence interval, the sums do not equal the growth and harvest volumes shown in Figure 3.)

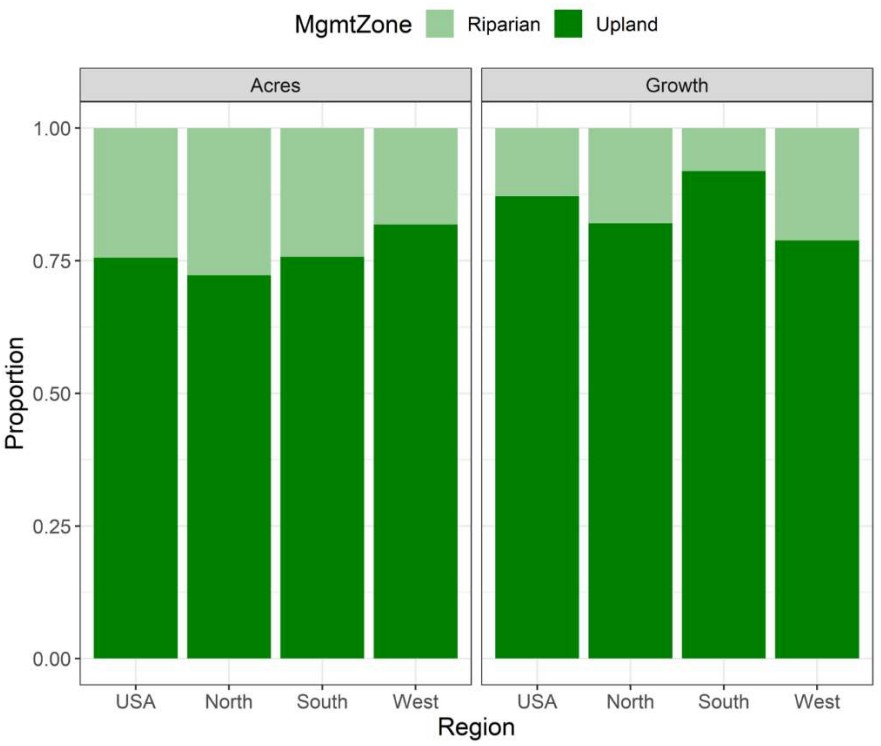

**Figure 6.** Proportion of FIA forest acres and growth volume by management zone.

**Table 12.** Proportion of FIA forest acres and growth volume by management zone.

| | Acres | | | | Growth Volume | | | |
|---|---|---|---|---|---|---|---|---|
| | USA | North | South | West | USA | North | South | West |
| Upland | 75.6% | 72.3% | 75.7% | 81.8% | 87.1% | 82.0% | 91.9% | 78.8% |
| Riparian | 24.4% | 27.7% | 24.3% | 18.2% | 12.9% | 18.0% | 8.1% | 21.2% |

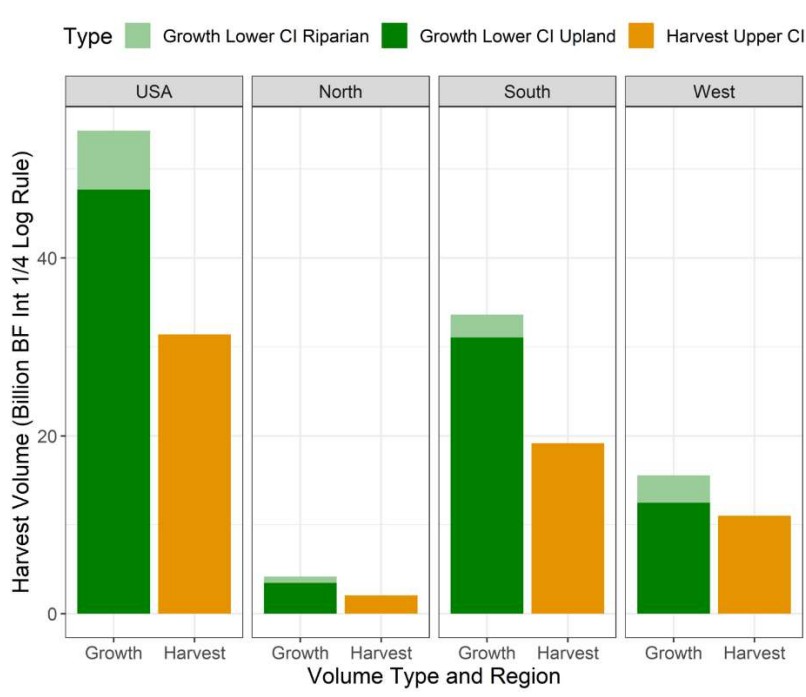

**Figure 7.** FIA estimated growth and harvest volume by management zone.

**Table 13.** FIA estimated growth and harvest volume by management zone.

| | Billion Board Feet (BBF) | | | | | | | |
| | USA | | North | | South | | West | |
| | BBF | Percent of Growth | BBF | Percent of Growth | BBF | Percent of Growth | BBF | Percent of Growth |
|---|---|---|---|---|---|---|---|---|
| Estimated Growth Riparian (Lower Confidence Interval) | 6.6 | | 0.7 | | 2.6 | | 3.1 | |
| Estimated Growth Upland (Lower Confidence Interval) | 47.7 | | 3.5 | | 31.1 | | 12.5 | |
| Estimated Harvest (Upper Confidence Interval) | 31.4 | 66% | 2.1 | 59% | 19.2 | 62% | 11.0 | 88% |

Note: Bisecting the data by zone increases standard error. Because we report the lower confidence interval, the sums do not equal the growth and harvest volumes shown in Figure 3.

Previously we estimated (Figure 3) that current harvests consume 56% of net growth. What we see in Figure 7 is that after deducting growth in riparian areas, harvests consume 66% of accessible growth. Now, as we did in the comparison between Figures 3 and 4, in Figure 8 and Table 14 we add incremental mass timber consumption to current harvesting and consider that compared with timber growth excluding riparian zones. Figure 8 is like Figure 4 but with growth split by riparian or upland (non-riparian). In this case, the elevated harvest volumes consume 82% of accessible growth. For the U.S., the lowest confidence interval for upland growth exceeds the upper confidence interval for harvests by 18% even when including the High-Volume estimate for mass timber incremental consumption. Looking regionally, in the North and South, excluding growth in riparian zones, we still grow more softwood timber than we use under the most optimistic mass timber forecast. Incremental consumption in the West could exceed growth when adjusted for the inaccessible riparian land, but we emphasize the word "could", as we are depicting the situation within the most conservative constraints.

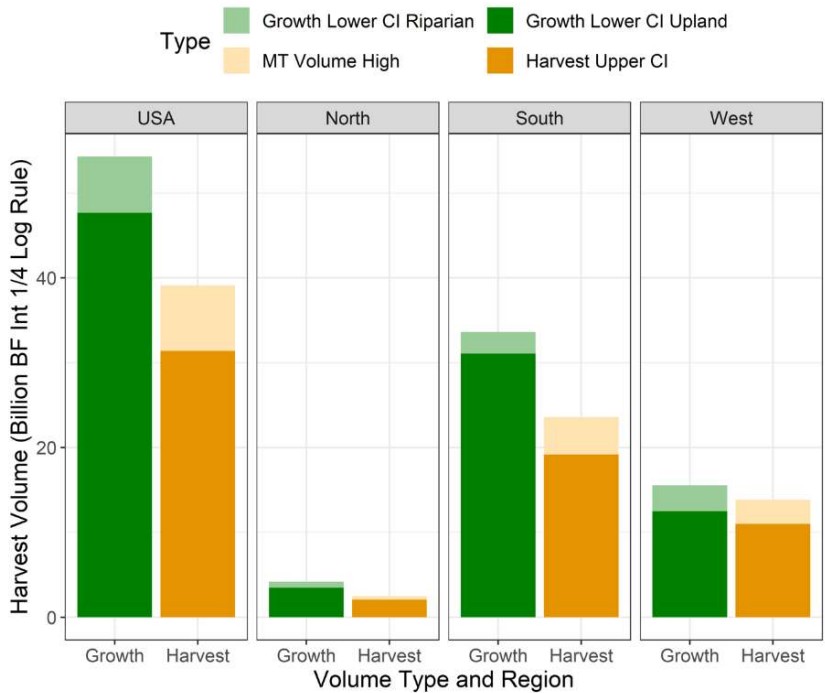

**Figure 8.** FIA estimated growth and harvest volume by management zone with projected mass timber demand.

**Table 14.** FIA estimated growth and harvest volume by management zone with projected mass timber demand.

| | Billion Board Feet (BBF) | | | | | | | |
|---|---|---|---|---|---|---|---|---|
| | USA | | North | | South | | West | |
| | BBF | Percent of Growth | BBF | Percent of Growth | BBF | Percent of Growth | BBF | Percent of Growth |
| Estimated Growth Riparian (Lower Confidence Interval) | 6.6 | | 0.7 | | 2.6 | | 3.1 | |
| Estimated Growth Upland (Lower Confidence Interval) | 47.7 | | 3.5 | | 31.1 | | 12.5 | |
| Estimated Harvest (Upper Confidence Interval) | 31.4 | 66% | 2.1 | 59% | 19.2 | 62% | 11.0 | 88% |
| Incremental Mass Timber Harvest (High Volume) | 7.7 | 16% | 0.4 | 13% | 4.4 | 14% | 2.8 | 23% |
| Excess Growth Considering Upland Only | 8.5 | 18% | 1.0 | 2.8% | 7.5 | 24% | −1.4 | 1.1% |

"Sustainability" is a complex concept that must consider site-specific and wide-ranging criteria. However, with 18% more growth using the lowest estimate over the highest estimated consumption, for the total U.S., the level of harvest is certainly sustainable from a timber volume perspective.

## 4. Discussion

Using USFS FIA data, incremental U.S. softwood timber harvests were projected that would be necessary to supply the Softwood Lumber Board's high-volume estimate of mass timber and light framing consumption in 2035. Growth in reserve forests and riparian zones was excluded, and low confidence intervals were used for growth estimates, compared with high confidence intervals for harvest and consumption estimates (i.e., the most conservative scenario). Results were considered for the U.S. in total and by three geographic regions (North, South, and West). In total, growth exceeds consumption including the incremental mass timber volumes. On a regional basis, growth exceeds consumption in the South and the North, but under these very conservative assumptions, consumption could exceed growth in the West region by around 10%.

To arrive at this conclusion, three related questions were investigated. First, what is the current rate of softwood harvesting relative to growth and reforestation? Using the lowest estimate of growth and the highest estimate of consumption, we find that current consumption is 56% of growth. This is not surprising, as timber inventories in the United States have been increasing for over fifty years.

What is the mass timber forecasted demand and volume relative to total annual consumption? Using the Softwood Lumber Board's most optimistic scenario, incremental lumber demand from mass timber use by 2035 would represent an increase of 17% over current softwood lumber consumption.

What is the forecasted rate of harvesting with projected mass timber demand relative to growth and reforestation? We find that the highest projected estimate of mass timber usage would be 12% of current growth, increasing total demand to 68% of growth. Recognizing that some growth occurs in areas that cannot be harvested (i.e., reserved lands and riparian zones), we reduced growth projections accordingly and find that the optimistic projections of consumption could reach 82% of the lowest level of projected growth. On a regional basis, and using these conservative estimates, only the Western regional demand could surpass growth.

Our results are not particularly surprising to anyone familiar with U.S. forest inventory data and the history of the U.S. timber supply. It is a well-known fact among forestry professionals, the forest products industry, and related academia that the U.S. timber

supply has been expanding for over half a century. However, this is not well known outside those circles. Cognitive dissonance related to a desire to use wood building products for their carbon storage capability but concerns about sustainability is understandable. The results of this research add more evidence that should help allay those concerns. Our analysis clearly shows that the United States can sustainably use more mass timber and reduce greenhouse gas emissions and embodied carbon in our built environment.

**Author Contributions:** Conceptualization, J.C., L.R. and K.W.; Data curation, J.C. and L.R.; Formal analysis, J.C. and L.R.; Funding acquisition, K.W.; Investigation, J.C. and L.R.; Methodology, J.C. and L.R.; Project administration, L.R. and K.W.; Software, J.C.; Supervision, L.R.; Validation, J.C. and L.R.; Visualization, J.C. and L.R.; Writing—original draft, J.C., L.R. and K.W. All authors have read and agreed to the published version of the manuscript.

**Funding:** This research was funded by the Binational Softwood Lumber Council (the BSLC), whose support is gratefully acknowledged. Design of the study, collection, analyses, and interpretation of data, writing of the manuscript, and the decision to publish the results was solely the work of the authors.

**Institutional Review Board Statement:** Not applicable.

**Informed Consent Statement:** Not applicable.

**Data Availability Statement:** This study used publicly available FIA data [10] and software [14]. Further questions regarding our use of the data can be addressed to the corresponding author.

**Conflicts of Interest:** The authors declare no conflict of interest.

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
