# Peer review of "Increasing Mass Timber Consumption in the U.S. and Sustainable Timber Supply"

_sustainability, doi:10.3390/su14010381_

Round 1

Reviewer 1 Report

The article discussed the possibility of replacing steel with mass timber, which is a practical and valuable study. The overall structure is clear, and the language and vocabulary are great.
But still need to make some minor adjustments, such as:
1- Can the author explain whether the calculation data used is the output of all timber or the volume of existing large timber?
2- In addition, is it possible to discuss how to reasonably use the existing mass timber and put forward some reasonable suggestions to make the article more complete.

Reviewer 2 Report

Increasing Mass Timber Consumption in the U.S. and Sustain-2 able Timber Supply

A brief summary

This topic is important because mass timber products are growing in popularity in U.S. as a substitute for steel and concrete, reducing embodied carbon in the built environment. Authors addressed the next question will rising demand for mass timber products result in unsustainable 54 levels of harvesting in coniferous forests in the United States?

The entire manuscript is well written and could be publish in sustainability journal in this version after minor revisions.

Dear author

Good job the manuscript is generally well written, and the topic is interesting for forestry literature.

Some comments, suggestions and observations are shown as general and specific comments below.

General comments:

The manuscript is well written. There are some comments and suggestion that could be used to improve the manuscript.

Please, improve the caption of figures according to the journal format.

Specific comments:

Abstract.

Congratulations, this section is professionally written. The written English is perfect.

Introduction.

Page 27. “a cross-laminated timber (CLT)” rather “a cross-laminated timber or CLT”.

L40. “CO2” rather “CO2”. Please, use subscript letter.

L58. Please use 42%.

L59. Please, use 380%.

L61. Please, make sure about Forest2Market.

L87. I think that table could be fit to window.

Is necessary the black color for main row?

Material and methods

L109. Percentage or percent?, which one is better?

L109. Please, mention the Table 3 before location. I think that Table 3 is missed in the text. The same for Table 4.

Could be put together the Table 3 and Table 4?, If yes, please reorganize the information in only one table.

Results

L158. Authors used “Softwood” Word.

L161. “… from 2010 to 2017…” Please, rewrite this sentence.

L163. “…volume high…”.

L168. Please, improve the caption of Figure 1, and do the same for all figures. Please, make sure about the organization of all Figures. There area a table combined with each figure.

Please take in mind if the figures could be improved.

L178. Please, improve the caption of Figure 2.

L185. “… Figure 2,…”

L198. Please, improve the caption of Figure 3.

L210. Please, improve the caption of Figure 4.

L211. Please, note the table in this line and not in the same way in other tables.

L226. “…Figure 5…”

L251. Please, improve the caption of Figure 6.

L252. The same.

L277. Please, improve the caption of Figure 7.

L299. Please, improve the caption of Figure 8.

Please, try to put all figures in the same format and size.

Discussion

God job.

L308. “…Softwood...”.

Conclusions

There is no conclusions section.

References

Well, done.

Reviewer 3 Report

This research sheds light on the future of mass timber products and consumption in the U.S. market.  The sustainability of U.S. timber supply is also discussed in this research. I recommend this manuscript be published in Sustainability journal. To improve the manuscript, please consider the following comments,
1-  In the Abstract, line 12, the authors should write what is the "USFS FIA" abbreviation refers to?
2-  This research studies mass timber, so there should be more details about it in the Introduction part, for example, mass timber products types as well as the most common woods used for their manufacturing particularly in the U.S. 
3- The history of mass timber products in the American marketplace is also important for readers, this should be added to the introduction part.
4- Lines 44-45 -What is the source of the information?
